# Exploring the Meta-level Reasoning of Large Language Models via a Tool-based Multi-hop Tabular Question Answering Task

## Abstract

Recent advancements in Large Language Models (LLMs) are increasingly focussed on "reasoning" ability, a concept with many overlapping definitions in the LLM discourse. We take a more structured approach, distinguishing meta-level reasoning (denoting the process of reasoning about intermediate steps required to solve a task) from object-level reasoning (which concerns the low-level execution of the aforementioned steps.) We design a novel question answering task, which is based around the values of geopolitical indicators for various countries over various years. Questions require breaking down into intermediate steps, retrieval of data, and mathematical operations over that data. The meta-level reasoning ability of LLMs is analysed by examining the selection of appropriate tools for answering questions. To bring greater depth to the analysis of LLMs beyond final answer accuracy, our task contains 'essential actions' against which we can compare the tool call output of LLMs to infer the strength of reasoning ability. We find that LLMs demonstrate good meta-level reasoning on our task, yet are flawed in some aspects of task understanding. We find that $n$-shot prompting has little effect on accuracy; error messages encountered do not often deteriorate performance; and provide additional evidence for the poor numeracy of LLMs. Finally, we discuss the generalisation and limitation of our findings to other task domains.

## 1 Introduction

Augmentation of Large Language Models (LLMs, (Brown et al., 2020; Radford et al., 2019; Devlin et al., 2019)) beyond text generation is now common, with *reasoning* ability across a range of tasks now a central feature (Dubey et al., 2024; Abdin et al., 2024). Reasoning is frequently benchmarked on question answering (QA) tasks which require decomposing a problem into smaller steps, which may involve mathematical reasoning (Cobbe et al., 2021; Hendrycks et al., 2021), commonsense reasoning over natural language facts (Talmor et al., 2019; Geva et al., 2021), or extracting tabular data Wu et al. (2024). Additionally, LLMs are no longer focussed only on the generation of natural language, but computer code and other structured outputs like function calls, as embodied in the tool-use paradigm (Mialon et al., 2023).

In this study, we discuss the *meta-* and *object-level reasoning* (Bundy, 1983) of LLMs using a multi-hop, data retrieval and arithmetic-based question answering task. Meta- and object-level reasoning are two modes originating with automated theorem proving and proof planning domain, yet have clear parallels with the reasoning discourse around LLMs. Meta-level reasoning encompasses the high-level planning task, the creation of a course of action for reaching a solution, and reasoning about the process of answering a question. While these tasks are commonly incorporated into a very general notion of 'planning' in the LLM community, when we focus on meta-level reasoning, we focus on one aspect, namely *the extent to which subcomponents of a system are correctly employed to achieve a specific goal.* Object-level reasoning encompasses the execution of the steps created by the meta-level process. This terminology is discussed in more detail in section 2, and serves as a framework against which we can evaluate and discuss the reasoning ability of different aspects of LLMs in a more structured manner beyond simple final answer accuracy.

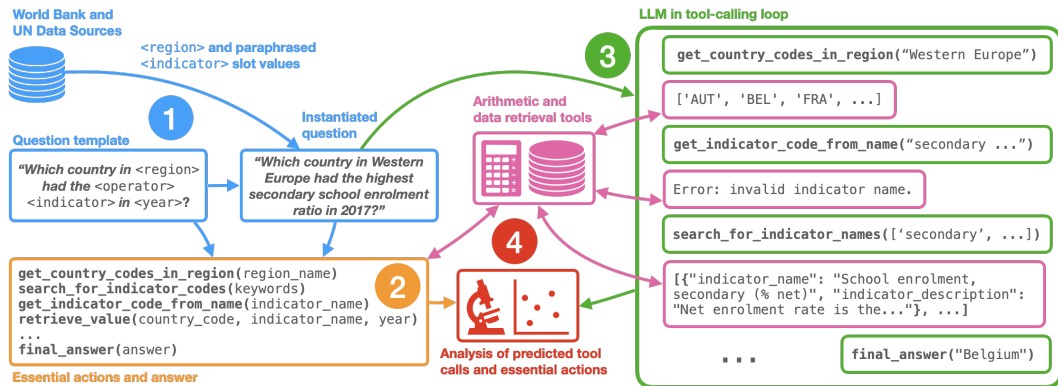

Figure 1: Overview of our question generation and evaluation process. **1** We instantiate question templates with slot values. **2** Using a hand-created templated sequence of required steps and the set of tools, we compute essential actions and answers. **3** Instantiated questions are passed to an LLM, which is held in a loop making tool calls which are executed and returned to the model. **4** The predicted set of tool calls are compared to the essential actions.

To investigate this reasoning ability in line with the current focus on application to multi-hop QA tasks requiring numeracy and planning, we design an evaluation environment comprising questions requiring meta-level reasoning (decomposition into intermediate steps) and object-level reasoning (retrieval of data from tabular sources and arithmetic operations)[1]. Our dataset concerns the values of World Bank indicator data for various regions, countries, and years, as illustrated in figure 1. However, the wider problem-solving task which we embody can generalise to other contexts and domains requiring high-level decomposition of a task into intermediate steps and low level execution of those steps, which may encompass data retrieval, symbolic and arithmetic operations, or informal natural language fact retrieval. We create 'essential actions' for each example in our dataset, which are a set of tool calls required to guarantee a correct answer, and against which we compare model-generated tool calls to infer meta-level reasoning ability. However, this is not a strict 'gold standard', single correct reasoning trace which we hold models to – we use this set of actions to analyse whether the model has satisfied the core aspects of the task. The aim of this work is not to design a system to maximise the performance of LLMs at our task, but rather to use the tool-use paradigm as an intermediate representation through which we can analyse the meta-level reasoning ability of LLMs. Consequently, we investigate the meta-level reasoning of off-the-shelf models without fine-tuning.

In parallel with our focus on the performance of LLMs at reasoning tasks, we are equally interested in the explainability and interpretability of the QA process, and this has informed the design of our environment. Similar to Wu et al. (2024), we are conscious of the relationship of research-based benchmarks to the use of LLMs as QA systems in industry, and are highly conscious of the proliferation of LLMs in commercial and professional settings, with a particular focus on QA. This motivation informs the design of our tool-calling evaluation loop, which allows us to not only inspect the reasoning process of LLMs via comparison with essential actions created in our dataset, but as a standalone feature enables highly interpretable outputs. To summarise our contributions, we:

- Create a multi-hop QA environment with 'essential actions' (§3).
- Evaluate LLMs in terms of final answer accuracy and meta-level reasoning ability (§5).
- Find that models are generally able to reason at the meta-level, selecting appropriate tools to achieve high accuracy.
- Analyse deficiencies in meta-level reasoning in terms of missed reasoning steps.
- Find that one- and three-shot examples of tool execution do not improve accuracy, but does reduce incorrect tool call frequency.
- Verify the limited numeracy of LLMs when we remove access to arithmetic tools.

---

[1]Our code is available at https://anonymous.4open.science/r/exploring-meta-level-reasoning-iclr-2026/

## 2 BACKGROUND

**Reasoning in LLMs**    In the context of LLMs, the term *reasoning* speaks to a systematic *problem-solving* or *decision-making* capability whereby inferences and conclusions are made based on available information Huang & Chang (2023). Reasoning may be subdivided into mathematical reasoning (Cobbe et al., 2021), symbolic reasoning (Wei et al., 2022), or commonsense reasoning (Bhargava & Ng, 2022); but is often linked with the task of breaking a problem down in to intermediate steps. Prompting models to explicitly generate intermediate steps to help solve problems improves performance at downstream tasks in zero- and few-shot settings (Wei et al., 2022; Wang et al., 2023; Kojima et al., 2022), while supervised fine-tuning also leads to performance gains on QA tasks (Talmor et al., 2019; Hendrycks et al., 2021). In this paper, we prefer to discuss meta- and object-level reasoning (which we overview in §2) not with the intention of superseding the above terms, but rather to provide a better structure to the discussion of the reasoning ability of LLMs.

**Tool-use**    Tool-use, is a paradigm in which LLMs can generate function calls to assist with task-solving (Wang et al., 2024; Schick et al., 2023), which are executed by external programs and the results returned to the model. They are typically used to alleviate intrinsic weaknesses in LLMs by improving arithmetic capability (Gao et al., 2023; Parisi et al., 2022) and real-time data retrieval through APIs, knowledge bases, and web search (Qin et al., 2024b;a; Lazaridou et al., 2022). An alternative interface to symbolic methods is the generation of code to perform a task (Drori et al., 2022; Chen et al., 2023; 2021).

**Existing Datasets**    A variety of datasets exist which examine the ability of LLMs to reason over questions requiring multiple intermediate steps of reasoning. GSM8k (Cobbe et al., 2021) and MATH (Hendrycks et al., 2021) focus on multi-hop mathematical reasoning tasks, while others focus on tasks which require reasoning over natural language evidence Talmor et al. (2019); Yang et al. (2018); Geva et al. (2021). While the ability of LLMs to interact with structured, tabular data is receiving significant attention Chen et al. (2020); Hegselmann et al. (2023); Zhu et al. (2021); Wu et al. (2024), they are not structured in a way that allows explicit analysis of the intermediate steps.

**Meta- and Object-Level Reasoning**    Meta- and object-level reasoning are terms associated with symbolic AI, particularly the automated reasoning and proof planning domains, yet they are highly relevant to the application of LLMs to QA. To help map these definitions to our task, we will first describe a range of examples of these two concepts to build up a picture of their meaning.

*Meta-level* reasoning refers to the reasoning about the representation of a theory, while the theory itself is at the *object-level* Bundy (1983). Bundy et al. (1979) use meta-level inference to control the search for a solution to mechanics problems, while object-level inference is used to compute the steps of the solution itself. Christodoulou & Keravnou (1998) describes meta-level reasoning as planning problem-solving strategies, controlling different *problem solvers* (object-level reasoning components), and notes the use of meta-level reasoning in adapting a strategy to new knowledge which may arise during computation. Aiello et al. (1991) describe meta-level reasoning as *reasoning about reasoning*, and note its use in driving search strategies and the modification of a system's own behaviour. In the context of an agent-based system, they distinguish meta- and object-levels by stating that agents' world knowledge is at the object-level, while meta-level knowledge governs links between agents. Genesereth (1983) distinguishes the actions of an AI system as *base-level* (or, object-level) and meta-level. Object-level actions achieve the program's goals, while meta-level actions decide which object-level actions to perform.

Nuamah et al. (2016) introduces a formalism for representing knowledge in a QA system consisting of attribute-value pairs. This formalism, developed in Nuamah & Bundy (2023), introduces additional attributes to the ⟨*subject*,*predicate*,*object*⟩ triple, which may be at the meta- or object-level. Object-level attributes encode the meaning of a factual statement, such as its *subject*, while meta-level attributes capture meta-information, such as the data source for a given fact. This example of meta- and object-level reasoning is applied to the FRANK system Nuamah & Bundy (2020), a symbolic reasoning framework applied to QA, and on which we base the design of our dataset. However, the symbolic meta- and object-level reasoning of the system requires a significant amount of hand-engineering, limiting generalisation to new operations or question types. In contrast, generalisation and reasoning are claimed strengths of LLMs, yet their ability to perform basic object-level

reasoning is poor. Given this context, our dataset was developed to embody the problem at which the FRANK system was targeted, and provide an environment in which we can compare the performance of LLMs at the meta- and object-level reasoning required by FRANK.

To summarise the above examples, meta-level reasoning corresponds to the high-level planning of a solution to a problem, the decomposition of a problem into intermediate steps. Reasoning at the object-level concerns the application of the subcomponents, including lower-level inferences such as mathematical operations or natural language deductions which are required to execute the intermediate steps. We find that this delineation of reasoning tasks provides meaningful detail and structure to the discourse and classification of the reasoning tasks embodied in multistep QA datasets on which LLMs are evaluated. Taking GSM8k as an example, it is commonly referred to as a mathematical reasoning benchmark, however upon analysis, the problems contained require meta-level reasoning to reason about the necessary intermediate steps for solving the problems, and object-level reasoning to correctly compute the values required by those steps. Our interpretation of the terms meta- and object-level reasoning is summarised below.

**Meta-level reasoning** *High-level planning.* With LLMs, we observe this via informal, natural language-based decomposition of a problem into sub-problems or intermediate steps, and create a structured manifestation using tool calls.

**Object-level reasoning** *Low-level execution.* With LLMs, this is demonstrated in the execution of intermediate steps created by the meta-level reasoning process. Execution of these steps may involve data retrieval, arithmetic, or even more informal processes such as natural language retrieval of facts.

## 3 OUR DATASET

### 3.1 QUESTION GENERATION

The dataset consists of 20 question templates which require meta-level reasoning to decompose a problem into intermediate steps, and object-level reasoning to perform arithmetic and data retrieval operations. While this style of question is not domain-specific, the content of our dataset is modelled around the World Bank Open Data platform[2], with answers derived from the values of a range of geopolitical indicators for different regions, countries, and years. Templates contain a variety of slots, such as *subject* and *region*, and, depending on the template, contain on the order of $10^3$ to $10^8$ possible values. Three examples are provided below, with the full twenty templates given in appendix C, and example slot values are shown in table 1. Question templates were hand-created to encompass a realistic range of tasks which may be performed over World Bank indicator data. Each template requires different combinations of a variety of elementary mathematical operations, such as summation, comparison, and ranking, in order to arrive at a final answer. In combination with these operations, questions also require data retrieval, which is facilitated by a set of tools which are called by supplying various arguments such as a country name, region, and year. Further detail is given in §4. Given the range of operations supported, different questions require answers of different types, such as lists, floats, integers, strings and boolean values. Each template contains between 2 and 4 hand-paraphrased forms.

**CountryThresholdCount** How many countries in `<region>` had a `<operator>` `<property>` than `<subject>` in `<year>`?

**RegionProportion** What proportion of the total `<property>` in `<region>` in `<year>` was contributed by `<subject>`?

**RegionPropertyChange** Which country in `<region>` had the `<operator>` increase in `<property>` between `<time_1>` and `<time_2>`?

The overall difficulty of the task is not high. It does not require domain-specific knowledge to decompose the problems or understand the necessary steps to achieve the answer. It is not designed to mislead models, contain 'trick' questions, or push models to the very limit of their ability. Rather, it is an instance of a more general style of problem which LLMs are frequently exposed to: requiring

---

[2]`https://data.worldbank.org/`

| Slot | Number available | Example(s) |
|------|------------------|------------|
| `<subject>` | 248 | *Ghana, France* |
| `<region>` | 22 | *Western Europe* |
| `<property>` | 94 | *Total population* |
| `<year>` | 20 | *2005, 2012* |
| `<operator>` | 2 | *Highest, lowest* |

Table 1: Summary of slot types for dataset questions.

intermediate reasoning steps, data retrieval, and arithmetic, and, to repeat, the dataset is designed to allow inspection and analysis of the reasoning process of LLMs beyond final answer accuracy.

## 3.2 Sourcing Data

The numeric data on which questions require consists of extracts from the World Bank's *featured indicators* downloaded from the World Bank Open Data API. Data provided in the API includes the indicator code (e.g., `AG.LND.CROP.ZS`), name (e.g., *Permanent cropland (% of land area)*) and a description. We use these fields to impose constraints on the 296 featured indicators for better question generation. We use indicator data for the years 2003-2023 to increase the proportion of available data, and we remove indicators which report 'normalised' values, e.g., *Agricultural land (% of land area)*. This reduces ambiguity – the presence of such phrases may mislead the model into normalising those values itself rather than using indicators with pre-normalised values. In this example, values for agricultural land area and country area may be retrieved separately and the percentage computed, rather than looking up the single normalised indicator. This is a valid approach, but not one built into our environment, although it could form the basis for a further study on reasoning. Similarly, we avoid question types which construct normalised values to avoid the same confusion. For questions which require information about a region, e.g., because the question concerns the average or maximum value across a set of countries, we use the United Nations Statistical Division's M49 standard[3] to classify countries as part of a regional set.

To improve the naturalness of our generated questions, we paraphrase indicator names from their ungrammatical initial forms using the indicator description. Three paraphrases were generated with GPT-4.1, using indicator description from the World Bank API. For example, *School enrolment, secondary (% gross)* is paraphrased to *Secondary school enrolment rate*, and *Rail lines (total route-km)* to *Railway route length*. The prompt used is provided in appendix B.

## 3.3 Generation of Solutions and Essential Actions

Answers to questions are created automatically using a template of function calls using the same set of tools which are provided to the models during generation. This enables evaluation of models' final answer accuracy on the questions as well as meta-level reasoning ability by comparing predicted, model-generated tool calls to this set of actions. There is not necessarily a single correct approach to answering each question, yet there *is* a core set of actions which *must* be taken in order to demonstrate proper meta-level reasoning over the tools provided, such as retrieving data. Hence, we refer to these sets of tool calls as 'essential actions'. This allows us to quantify a range of intuitive notions of performance, with high similarity between predicted tool calls and essential actions indicating efficient, strong meta-level reasoning to low similarity indicating poor ability.

## 3.4 Unanswerable Questions

Data is not available for all countries, years, and indicators, meaning that some questions inevitably cannot be answered. As such, in the dataset we distinguish between *answerable* and *unanswerable* questions. Answerable questions contain only *full* data availability, indicating that there are no missing values whatsoever in the data relevant to the question. Missing data naturally indicates that critical information is not present to enable the model to compute the answer. A third mode, *partial* data availability means that enough data exists for the question to be answerable, but not all fields are

---

[3]https://unstats.un.org/unsd/methodology/m49/

---

**Algorithm 1** Evaluation of an example from dataset

---

**Input:** Question $q$, model $M$, tools $T$
**Output:** Final answer $a$, predicted tool calls $C$

$S \leftarrow \{\text{system prompt, user question } q\}$ ;           // initialise dialogue state
$C \leftarrow []$ ;                     // initialise predicted tool call sequence
action $\leftarrow$ None
**while** $a = None$ **do**
    $T_{pred} \leftarrow M(S)$ ;       // sequence of tool calls produced from state $S$
    **forall** $t \in T_{pred}$ **do**
        **if** $t \in T$ **then**
            $C \leftarrow C \parallel [t]$ ;          // append tool call to predicted sequence
            $o \leftarrow t()$ ;                    // execute tool and obtain output
            $S \leftarrow S \cup \{o\}$ ;                   // return tool output to model
        **else if** $t = FinalAnswer$ **then**
            $a \leftarrow$ model's final answer

**return** $(a, C)$

---

available. For example, when retrieving values for all countries in a given region, data may not be available for all countries. Yet, it is still possible to perform summation or averaging over such data. While we do not incorporate such as setting in our study, this mode offers an interesting platform for further analysis of meta-level reasoning.

## 4 EXPERIMENTAL SETUP

**Tool Creation**  22 tools were created to allow the models to perform object-level reasoning including mathematical operations and data retrieval. 13 are elementary arithmetic operations, and are immediately applicable to other domains and evaluation scenarios. Additionally, seven data retrieval tools allow models to retrieve local World Bank data stored in CSV files. While these are designed to access World Bank data, they are not domain-specific in their overall functionality, and complementary tools could easily be created to perform object-level reasoning processes in different scenarios. A *think* tool allows a model to generate natural language text to guide their reasoning; and a *final answer* tool aids answer parsing. A full overview is provided in appendix 3.

The data retrieval tools include a `search_for_indicator_names` tool, which interfaces a list of indicator names and descriptions. The `get_indicator_code_from_name` tool returns the relevant code for an indicator name, needed for accessing data with the `retrieve_value` tool. Similar tools exist for retrieving a country code, or the country codes belonging to region. Tools return errors if used incorrectly, for example, if a non-existent indicator code is used in `retrieve_value`, and are used to examine if the model is able to recover from mistakes.

While each question template requires a different approach, all templates require data retrieval and arithmetic operations, and some patterns are found across templates. A question may require the following steps, as initially outlined in 1. Beginning with the question, e.g., "*Which country in Western Europe had the highest secondary school enrolment in 2017?*" models should search for available indicators using keywords from the question, such as *secondary*, *school*, and *enrolment*. The correct indicator should be inferred from the output of this tool, and its code retrieved. Country codes for the country in question, or, if required, the country codes for a given region (e.g., *Western Europe*) should be retrieved. Numeric data retrieval will follow, using country codes, indicator codes, and the year, as all data is stored in separate CSV files and indexed by country code and year. Arithmetic tools are then required for operating over that retrieved data in order to provide the final answer to the question – in this example case, a simple `max` operation.

**Tool Calling Loop**  We prompt the model with a Chain-of-Thought- and ReAct-style approach, instructing the model to create a step-by-step plan for answering the question, breaking down the question into a series of actionable steps to be executed using tools, and encouraging the model to take regular 'thinking' steps. Full details of the prompts used are provided in appendix A.

Rather than a simple SQL or code generation approach, which would not allow for the reasoning over intermediate results from intermediate steps of the QA process, we hold the model in a loop in which tool calls are executed until the 'final answer' tool is called. This process is illustrated in algorithm 1. Models were run with recommended generation parameters from model providers.

## 5 RESULTS

In this section, we provide an overview of the meta-level reasoning capability of models, which we approach using a modified precision and recall, comparing predicted tool calls to essential actions. Precision and recall allow robust assessment of the model's meta-level reasoning beyond simple final-answer accuracy by rewarding the model for producing correct tool calls, while penalising incorrect or irrelevant tool calls. Additionally, because the essential actions are a set of discrete components, we avoid a brittle single 'gold standard' comparison. There are not multiple competing, valid reasoning approaches to answering questions – the only minor variations are to be found where, for example, models may perform an `add` call followed by a `divide` call instead of simply calling the `mean` tool in a single step. While the same result is achieved, this results in a minor correction to precision and recall. This correction reflects the intuition behind our modified precision and recall – applying a minor penalty for not selecting the correct tool is what we wish to show. Consequently, when a model generates numerous tool calls, many of which may be repeated multiple times, the model will be more heavily penalised for demonstrating understanding of the correct approach.

Before computing precision and recall, post-processing is performed over predicted tool calls to credit the model for tool calls semantically equivalent to essential actions. First, we normalise all `less_than` tool calls to `greater_than` calls, with values reversed, because all essential actions comparisons are formatted as greater-than comparisons. Any `search_for_indicator_names` call which returned the correct indicator name is counted as a true positive. For tools which take a list of arguments, e.g., `add`, we count any call as a true positive if the values are correct. We penalise repeated tool calls of the same arguments by recording only one instance of a tool call as a true positive, and the rest as false. Finally, we do not include `think` or `final_answer` calls in our calculation of true or false positives.

**Precision and Recall** Higher accuracy and precision indicates a that a model is able to grasp the meta-level reasoning requirement well, selecting appropriate tools to complete subcomponents of the question answering process to efficiently arrive at an answer. Lower precision indicates that a model has made tool calls which are irrelevant or unnecessary, and are an indication of weak meta-level reasoning. Similarly, recall indicates the proportion of essential actions that the model took. Higher recall values show that models performed a high proportion of essential actions, while lower values indicate that models performed actions implicitly or simply ignored steps.

With reference to table 2, accuracies of approximately 0.6-0.8 were observed across the range of models evaluated, suggesting that models are able to demonstrate the meta-level reasoning requirements across a proportion of our task. High precision was frequently observed as in the case of Qwen 3 32B, indicating that models possessed a strong understanding of the process by which the tools should be used to answer questions. One cause of lower precision is that models will attempt a `get_indicator_code_from_name` call with a non-existent indicator name. Only after receiving an error from this will they call `search_for_indicator_names` and resume the correct process. Valid approaches which result in lower precision include performing `add` and `divide` calls rather than using the `mean` tool for questions which require averages. Poor performance of Llama 3.3 is derived from the hallucination of indicator codes, which propagates to many incorrect `retrieve_value` calls, and eventually the incorrect answer.

As with precision, high recall values are consistently observed, but the poorer performance results from models assuming performing basic arithmetic operations without the use of tools. When recall is very low, this is usually an indicator that model has hallucinated a country or indicator code, leading to a number of incorrect `retrieve_value` calls. While such implicit operations may sometimes be correct, models are explicitly prompted to use a tool to complete a task if one is available, and so such failures contribute to poor meta-level reasoning. We did not observe hallucination data values – models always attempted to use the retrieval tool.

| Model | $n$ | Err. | Acc. | Precision | Recall | Model | $n$ | Err. | Acc. | Precision | Recall |
|---|---|---|---|---|---|---|---|---|---|---|---|
| Llama 3.3 70B Instruct | 0 | 0.34 | 0.39 | 0.33±0.39 | 0.28±0.37 | Mistral Small 3.1 24B | 0 | 0.05 | 0.77 | 0.88±0.21 | 0.88±0.21 |
| | 1 | 0.28 | 0.37 | 0.40±0.40 | 0.30±0.37 | | 1 | 0.06 | 0.79 | 0.88±0.21 | 0.87±0.21 |
| | 3 | 0.20 | 0.28 | 0.30±0.38 | 0.19±0.30 | | 3 | 0.06 | 0.77 | 0.87±0.22 | 0.85±0.23 |
| Qwen 3 4B | 0 | 0.67 | 0.60 | 0.76±0.32 | 0.66±0.31 | Qwen 3 14B | 0 | 0.19 | 0.58 | 0.85±0.26 | 0.72±0.28 |
| | 1 | 0.50 | 0.60 | 0.77±0.34 | 0.64±0.31 | | 1 | 0.23 | 0.61 | 0.83±0.28 | 0.71±0.28 |
| | 3 | 0.56 | 0.53 | 0.67±0.38 | 0.55±0.35 | | 3 | 0.44 | 0.57 | 0.78±0.29 | 0.67±0.29 |
| Qwen 3 30B-A3B | 0 | 0.24 | 0.68 | 0.85±0.27 | 0.71±0.27 | Qwen 3 32B | 0 | 0.34 | 0.84 | 0.90±0.21 | 0.81±0.22 |
| | 1 | 0.18 | 0.67 | 0.87±0.25 | 0.74±0.25 | | 1 | 0.24 | 0.86 | 0.91±0.20 | 0.81±0.22 |
| | 3 | 0.22 | 0.66 | 0.87±0.25 | 0.73±0.25 | | 3 | 0.19 | 0.84 | 0.91±0.19 | 0.79±0.21 |
| GPT 4o Mini | 0 | 0.52 | 0.68 | 0.67±0.30 | 0.74±0.31 | GPT 4.1 Mini | 0 | 0.53 | 0.70 | 0.79±0.23 | 0.81±0.20 |
| | 1 | 0.40 | 0.64 | 0.63±0.34 | 0.67±0.34 | | 1 | 0.37 | 0.70 | 0.81±0.21 | 0.81±0.20 |
| | 3 | 0.40 | 0.64 | 0.64±0.33 | 0.67±0.33 | | 3 | 0.36 | 0.70 | 0.82±0.23 | 0.81±0.20 |

Table 2: Results on a sample of 400 questions (20 per type), with access to all tools, on the answer-able split of the dataset with full data availability. Each model above was evaluated in zero-, one-, and three-shot settings. **Err.** indicates the proportion of outputs which contained at least one tool call resulting in an error, and this is reported alongside with final answer accuracy (**Acc.**), and our modified precision and recall (± one standard deviation).

Across the models evaluated, **model size is not a guaranteed indicator of performance**, with Qwen 3 4B outperforming Llama 3.3 70B, although performance did improve within the Qwen 3 family as model size increased. Qwen 3's *reasoning/thinking* mode – in which paragraphs of text are generated to guide its approach to answering the question – is likely the primary cause of such high performance with respect to model size, but additional experiments are required to verify this. Llama 3.3 8B Instruct and Llama 3.2 3B Instruct were also evaluated, but performance was close to zero across our metrics, and so were not included.

$n$**-shot Prompting** $n$-shot prompting aids performance by providing examples of expected out-puts. For example, when paraphrasing indicator names in §3.2, we could have provided example paraphrases to improve results[4]. Providing example reasoning traces would exhaust models' context windows and weaken the focus of our study in examining the off-the-shelf meta-level reasoning of LLMs, so we provided models with $n$ examples of the inputs and outputs of each tool using randomly generated arguments. **Our implementation of $n$-shot prompting did not increase performance** across the models evaluated, in some cases causing a decrease in performance by 10 percentage points in the case of Llama 3.3 according to the results in table 2. Large performance decreases were not common – example tool calls had little effect on overall performance, suggesting that the mechanics of the tools was not limiting the meta-level reasoning of models. However, across some evaluations, $n$-shot prompting reduced the proportion of examples which contained a tool call that resulted in an error, as in the case of Llama 3.3 and Qwen 3 4B and 32B. While most cases resulted in this reduced or maintained error rate, there is one outlier in Qwen 3 14B, with over twice as many examples containing errors when incorporating three-shot prompting.

**Error Messages** A key aspect of meta-level reasoning is the productive use of failure, so we exam-ine the frequency of cases where models were able to achieve the correct answer despite incorrectly using a tool. If a tool was called with incorrect arguments, an error message is returned to the model explaining the error. Figure 2a shows the influence of a faulty tool call and resulting error message on the likelihood of the correct answer being found. Consistent behaviour is not observed across models: while Qwen 3 4B, 32B and GPT models saw accuracy maintained or even increased in the presence of an error, intermediate Qwen 3 model sizes do not demonstrate similar results. Llama 3.3 70B, which performed poorly, benefitted from the error messages as demonstrated by a higher accuracy when an error was made, suggesting that while its pre-training on tool use may have been poorer, there may be stronger meta-level reasoning than the results in table 2 indicate.

---

[4]We did not choose this approach however, as generated paraphrases were already of sufficient quality without.

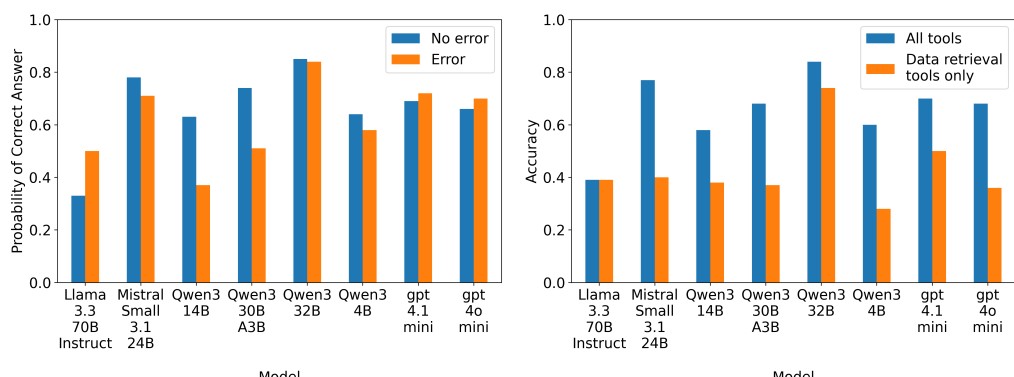

(a) Effect of the presence of an error on final answer accuracy.

(b) Comparison of zero-shot final answer accuracy using all tools, and only data retrieval tools.

Figure 2: Comparison of experimental results: (a) effect of error presence on final answer accuracy, and (b) zero-shot accuracy with all tools vs. data retrieval only.

**Object-level Reasoning** We evaluated the object-level reasoning ability of LLMs by restricting models to only data retrieval, requiring all mathematical operations to be performed in the standard text generation output. Despite improving performance of LLMs on arithmetic tasks, performance was degraded by the absence of dedicated symbolic functions, as shown in figure 2b. While only 10 percentage points lower in the case of Qwen 3 32B, increasing model size is not guaranteed to fix this weakness. This corroborates existing results that LLMs remain severely limited at basic mathematical tasks, demonstrating that external symbolic functions are still essential for such tasks.

## 6 CONCLUSION AND FURTHER WORK

In this study, we evaluated one aspect of the meta-level reasoning of LLMs via a multi-hop tabular QA task. We created a set of tools comprising elementary mathematical operations and data retrieval to perform object-level reasoning, and studied the meta-level reasoning of LLMs by comparing their tool selection behaviour to a set of essential actions required to rigorously answer each question. The dataset construction, and to a greater extent the construction of our evaluation, are applicable to wider studies of the reasoning ability of LLMs with respect to multi-hop QA.

We observed high accuracy and consequently infer strong meta-level reasoning by some models via high precision and recall scores, and suggest that reasoning performance is dependent on reasoning-oriented and tool-use fine-tuning. Even when primed with three-shots of example tool-use, we did not observe improved results, although we did observe a lower incidence of error-inducing tool calls. Error messages were used productively by five of the eight models evaluated, demonstrated by a marginal change in accuracy when errors were encountered, indicating that models were able to to understand why an error was made and re-execute a given step. Finally, we confirmed the necessity of symbolic functions for object-level reasoning by observing substantial decreases in accuracy in the absence of dedicated arithmetic tools.

To return to our introductory words on the topic of LLMs and reasoning: reasoning is a multi-faceted concept which, in this work, we offer a more structured analysis of one aspect of the reasoning ability of LLMs. Our work indicates that LLMs show good meta-level reasoning ability, though further study is necessary to make a more general comment across a range of problem domains and difficulties. Our environment opens many of these directions for further investigations of meta-level reasoning, such as examining reasoning under uncertainty and re-planning. Similarly, broadening the actions contained within essential action sets would allow for multiple reasoning paths would allow for richer evaluation; exploring a wider variety of problem contexts within our framework would confirm the generalisability of our results; and evaluating more challenging scenarios would function to explore the limitations of LLMs' conceptual understanding.

## 7 ETHICS STATEMENT

No ethical concerns were raised over the course of this work.

## 8 REPRODUCIBILITY STATEMENT

The source code for our dataset generation, evaluation environment, analysis and results is available at `https://anonymous.4open.science/r/exploring-meta-level-reasoning-iclr-2026`. Generation of the dataset, including essential actions, is found in `frankenstein/`, with question templates in `templates/` and tools implemented in `tools/`. Raw World Bank data is fetched and stored in `resources/`. The sample of the dataset used in evaluation is available at `dataset/answerable-full.jsonl`. `eval/` contains scripts for evaluating models on the dataset, including LLM outputs in `eval/runs`. The core algorithm as shown in algorithm 1 is implemented in the `loop` method of the `Runner` class in `eval/runner.py`. Results and analysis scripts which inform section 5 are also found in this directory. *Please note that in the 'supplementary material' submitted via OpenReview, most of the files in* `eval/runs` *have been removed to bring the file size under the 100Mb limit, but these are all present in the anonymised link above.*

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

## A  EXPERIMENTAL LOOP PROMPTS

### A.1  BASE SYSTEM

This prompt is included in all experiments.

> *You are a helpful assistant tasked with answering questions that require multiple intermediate steps of reasoning to arrive at a final answer.*
> *The questions involve using World Bank data for various countries and indicators.*
> *The question cannot be answered in a single step, so you must break it down into smaller tasks, and use the results of each step to inform the next step.*
> *Create a step-by-step plan to answer the question, and then execute each step of that plan to arrive at the final answer.*
> *If you need to, take the time to think through the problem and plan your approach before acting.*
> *To help me parse your answer, only provide the answer itself (e.g., the number, list, string, or boolean value) as your answer. Do not include any additional text or explanations. Do not perform any rounding or formatting of the answer.*

### A.2  BASE TOOL-USE

This prompt, clarifying the tool-use process, also appears in all experiments.

> *You have access to a set of tools to help you answer the question:*
> *Pay attention to the tool names, arguments, descriptions, and the types of outputs they return, and think carefully about how to use them to solve the problem.*
> *If there is a tool available that can help you with the next step, you must use it rather than trying to solve the problem without it.*
> *Do not format tool calls inside message content, instead, create them as dedicated tool calls in the 'tool_calls' field of the message.*
> *I will execute tool calls that you provide. You can use multiple tools in one step, but make sure you follow the correct format.*
> *Use the results of each tool call to inform your next step. **Passing tool calls as arguments to other tool calls is not allowed.** Instead, execute each tool call separately and use the results to perform subsequent calls – I will not execute nested tool calls.*
> *If a tool call fails, use the error message to help you debug the issue, re-plan, and*

*try again if possible.*

*Only provide the answer itself (e.g., the number, list, string, or boolean value) as your answer. Do not include any additional text or explanations. Do not perform any rounding or formatting of the answer.*

***You must create a 'final_answer' tool call to return your final answer - I will not be able to parse your answer from message content.***

### A.2.1 ALL TOOLS

When all tools are provided to the model, the following prompt is appended.

*The tools you have access to are below:*
*¡list of tool signatures with tool name, description, and arguments¿*

### A.2.2 DATA TOOLS-ONLY

When only data retrieval tools are made available to the model, the following prompt is instead appended.

*The tools you have access to are below:*
*¡list of tool signatures with tool name, description, and arguments¿*
*These tools allow you to access World Bank indicators and retrieve data for specific countries, indicators, and years. Use them to fetch relevant data to answer the question.*
*However, you must **perform any necessary arithmetic manually**, without tool support for computation. If the answer requires calculations (e.g., summation, averages), you must compute these yourself based on the retrieved data.*

## B INDICATOR PARAPHRASING PROMPTS

The following prompt was used to paraphrase original World Bank indicator names.

*You are a helpful assistant that paraphrases World Bank indicator names using the context provided in the additional description.*
*Return exactly three clear, concise **noun phrases** that faithfully represent the meaning of the original indicator name. Output them as a semicolon-delimited list.*
*These noun phrases will be inserted into questions like:*
*- "Which country in Eastern Europe had the highest ¡paraphrased indicator name¿ in 2020?"*
*- "Was the average ¡paraphrased indicator name¿ in Northern America higher or lower than the value for Ghana in 2020?"*
*- "What was the ¡paraphrased indicator name¿ in 2020 for the country with the highest value in South Asia?" - "Did ¡country¿ have a higher ¡paraphrased indicator name¿ than ¡other_country¿ in 2020?"*
*Write the paraphrases **as if a person were using them to ask a question like the ones above**. Make them sound **natural and conversational**, like something someone would realistically say or hear, without compromising technical accuracy.*
*Follow these guidelines:*
*- Make all outputs concise, grammatical, easy to understand and **suitable for inserting into questions** like these.*
*- Compress the phrase into the **shortest possible form** while retaining the meaning.*
*- Do not use the words **total** or **average** in the paraphrase as this will interfere with the grammar of the wider questions.*
*- Include bracketed elements, e.g., "(% of GDP)" as natural language phrases, such as "as a percentage of GDP". - **Do not include units of measurement**, e.g., "in US dollars", or "in TEUs".*

*- Avoid embellished and abstract language, or esoteric terms. If an indicator name is very simple (e.g., 'rural population', 'net migration', 'surface area'), use that as one of the three paraphrases.*
*- \*\*Only capitalize proper nouns or acronyms\*\*. Even though these are noun phrases, they will be inserted into the middle of sentences.*
*- Use the additional description only to \*\*clarify meaning\*\*, not to add new information.*
*- To repeat, paraphrases should be \*\*noun phrases\*\*. Start the phrase with something like 'count of', 'number of', 'percentage of', 'area of', 'rate of' if you are not sure how to begin.*
*Reminder: preserve the meaning of the original indicator name; shorten as much as possible; and do not use unusual phrasing.*

## C   QUESTION TEMPLATES

The full list of twenty templates are provided below. Paraphrased question forms are not shown.

**AverageChange** What was the average yearly change in `<property>` for `<subject>` between `<year_a>` and `<year_b>`?

**AverageProperty** What was the average value of `<property>` in `<region>` in `<year>`?

**AveragePropertyComparison** Was the `<property>` of `<subject>` `<operator>` than the average value for `<region>` in `<year>`?

**CountryPropertyComparison** Did `<subject_a>` have a `<operator>` `<property>` in `<year_a>` than `<subject_b>` had in `<year_b>`?

**CountryThresholdCount** How many countries in `<region>` had a `<operator>` `<property>` than `<subject>` in `<year>`?

**PropertyOfSubject** What was the value of `<property>` for `<subject>` in `<year>`?

**PropertyRatioComparison** Was the ratio of `<property>` for `<subject_a>` to `<subject_b>` in `<year>` `<operator>` than some threshold?

**RankChange** Did the rank of `<subject>` in `<property>` in `<region>` change between `<year_a>` and `<year_b>`?

**RegionAverageComparison** Did `<region_a>` have a `<operator>` average `<property>` than `<region_b>` in `<year>`?

**RegionComparison** Which country in the region of `<region>` had the `<operator>` `<property>` in `<year>`?

**RegionComparisonResult** For the country in `<region>` that had the `<operator>` `<property>` in `<year_2>`, what was its value in `<year_1>`?

**RegionPropertyChange** Which country in `<region>` had the `<operator>` change in `<property>` between `<year_a>` and `<year_b>`?

**RegionPropertyRatio** What was the ratio of `<property>` values in `<region>` in `<year>`?

**RegionProportion** What proportion of the total `<property>` in `<region>` in `<year>` was contributed by `<subject>`?

**RegionProportionChange** Was `<subject>`'s share of the total `<property>` in `<region>` `<operator>` in `<year_a>` than it was in `<year_b>`?

**RegionRangeComparison** Did `<region_a>` have a `<operator>` range of values for `<property>` than `<region_b>` in `<year>`?

**SubjectPropertyChange** Did `<subject>` have a `<operator>` change in `<property>` between `<year_a>` and `<year_b>`?

**SubjectPropertyRank** What was the rank of `<subject>` in `<property>` in `<region>` in `<year>`?

**TopNTotal** Which `<n>` countries in `<region>` had the `<operator>` total `<property>` in `<year>`?

**TotalProperty** What was the total value of `<property>` in `<region>` in `<year>`?

# D   FULL TOOLSET

The full set of tools that models have access to is shown in table 3. The first section is data retrieval tools, the second arithmetic, and third 'utility'.

19

| Name | Description | Arguments |
|------|-------------|-----------|
| search_for_indicator_names | Retrieve indicator names and descriptions that match the given keywords. | keywords: A list of keywords or a string to search for. |
| get_country_code_from_name | Get the three-letter country code from a country name. | country_name: The name of the country to get the code for. |
| get_country_name_from_code | Get the country name from a three-letter country code. | country_code: The three-letter country code to get the name for. |
| get_indicator_code_from_name | Get the indicator code from an indicator name. | indicator_name: The name of the indicator to get the code for. |
| get_indicator_name_from_code | Get the indicator name from an indicator code. | indicator_code: The code of the indicator to get the name for. |
| get_country_codes_in_region | Get the list of country codes in a given region. | region: The region to get the countries for. |
| retrieve_value | Return the value of an indicator for a country at a given year. | country_code: The three-letter country code; indicator_code: The indicator code; year: The year to look up. |
| add | Add a list of numbers. | values: A list of numbers to add. |
| subtract | Subtract value_b from value_a. | value_a: The first number; value_b: The second number. |
| greater_than | Check if value_a is greater than value_b. | value_a: The first number; value_b: The second number. |
| less_than | Check if value_a is less than value_b. | value_a: The first number; value_b: The second number. |
| multiply | Multiply a list of numbers. | values: A list of numbers to multiply. |
| divide | Divide two numbers. | value_a: The first number; value_b: The second number. |
| mean | Calculate the mean of a list of numbers. | values: A list of numbers to calculate the mean for. |
| maximum | Return the maximum of a list of numbers. | values: A list of numbers. |
| minimum | Return the minimum of a list of numbers. | values: A list of numbers. |
| count | Count the number of non-None elements in a list. | values: A list of values to count. |
| rank | Return the 1-based rank of query_value in values sorted descending. | values: A list of numbers; query_value: The value whose rank is to be determined. |
| sort | Sort a list of numbers. | values: The list of numbers to sort. |
| index | Return the 0-based index of query_value in values. | values: List of values; query_value: The value to find the index for. |
| think | Record a thought or plan for the next step. | thought: A string describing your plan or reasoning. |
| final_answer | Submit your final answer. | answer: The answer to the question. |

Table 3: Metadata for tools.