# OpenReview forum: "Exploring the Meta-level Reasoning of Large Language Models via a Tool-based Multi-hop Tabular Question Answering Task"
_ICLR.cc/2026/Conference — Submitted to ICLR 2026_

### Official Review · Reviewer_YaQy · 2025-10-16

**Soundness:** 2
**Presentation:** 1
**Contribution:** 2
**Rating:** 2
**Confidence:** 3

**Summary:**

The authors propose a new QA system to evaluate the meta-reasoning capability of LLMs by supervising the intermediate reasoning chain by examining the correctness of core tools selected.

**Strengths:**

- This paper proposes a new method to validate the meta-reasoning ability of LLMs.
- The paper tests several models using their benchmark.

**Weaknesses:**

- Table 2 exceeds the width of the paper template, which should not appear.
- The benchmark needs human annotation to ensure its fairness and reasonableness. However, no relevant statistics are provided.
- The baseline used are mainly prompt engineering. More updated baselines should be included.

**Questions:**

See above.

---

> ### Author Response · Authors · 2025-11-24
>
> Hello, thank you for your review.
>
> Given the extent of your described weaknesses, we can't help but feel that the overall rating of the paper is entirely unjustified. None of the points raised engage with the central criteria for evaluating a scientific contribution: whether our hypotheses are clearly stated, whether the methodology is sound, and whether the experimental results support or contradict the claims being made, criteria which we believe our work does satisfy. You do not point to any specific technical issue, methodological error, or misinterpretation of results.
>
> Your listed weaknesses are not entirely irrelevant. For example, we are happy to adjust the formatting of the table. Human annotation that the questions are reasonable is a good idea. Yet your last point on baselines is very vague, and mapping your intended meaning onto the fine detail of our study in order to discuss the point further is hard to do without further information.
>
> A rating of two based solely on the described weaknesses feels very severe and disproportionately low. None of these points you raise suggest fundamental, serious, scientific flaws with the contribution itself, and so we invite a serious reconsideration of our work.

---

> > ### Comment · Reviewer_YaQy · 2025-11-26
> >
> > Thank you for your feedback. However, I don't see the contribution of the paper that could be accepted.
> > - I strongly recommend that the authors look carefully at the organization of the paper. A table whose width exceeds the template should not happen in such a conference. Also, your table on page 19 is completely out of order.
> > - You are creating the benchmark and testing on this benchmark to draw your final conclusion. However, for such a procedure, the effectiveness and quality of the benchmark must be ensured to preserve the accuracy of the conclusion. A benchmark without human annotation results in unfairness and skepticism.
> > - All of your experiments are based on prompt engineering, including direct prompt and icl. At least, you should try more updated prompt engineering methods, like the agentic framework or ToT. The experiment is too limited, and the analysis is lacking.
> > - The tense usage throughout the paper is inconsistent. For example, in "If a tool was called with incorrect arguments, an error message is returned to the model explaining the error," you shift from the past to the present tense unnecessarily. The presentation requires significant improvement.
> > Overall, I still strongly recommend rejecting the paper.

---

### Official Review · Reviewer_Qa2K · 2025-10-29

**Soundness:** 2
**Presentation:** 2
**Contribution:** 2
**Rating:** 2
**Confidence:** 4

**Summary:**

This work proposed a tabular benchmark to evaluate meta-level reasoning of LLMs via a multi-hop tabular QA task. It includes human-defined tools and question templates on World Bank indicator data. Several open source or proprietary are evaluated via zero-shot and few-shot methods, showing gap on the proposed benchmark.

**Strengths:**

1. Problem framing: Clear split between object-level and meta-level, which is useful for capability-wise diagnosis and future extensions.

2. Beyond accuracy, the paper reports precision/recall (±1σ) and an Err. rate (share of outputs with at least one tool-call error), capturing both process robustness and final correctness.

3.  Comparisons across 0/1/3-shot, error vs. no-error conditions, and tool-set ablations (all tools vs. data-retrieval only) reveal more perspectives.

4. Clarity. Method and experimental setup are straightforward and easy to follow.

**Weaknesses:**

1. The benchmark relies on templated prompts and relatively narrow tools APIs, which is too simple compared to current practice, e.g., richer coding/spreadsheet/enterprise interfaces such as in SheetBrain: A Neuro-Symbolic Agent for Accurate Reasoning over Complex and Large Spreadsheets (https://arxiv.org/abs/2510.19247).

2. The study omits stronger/ newer models (e.g., GPT-5, Gemini 2.5 Pro) and larger mainstream open models (e.g., newer Qwen3 variants), making it hard to gauge the field’s current frontier.

3. Insufficient error study. Beyond aggregate Acc./Precision/Recall and brief text, there is no systematic analysis of agent failure causes across models, nor full examples that illuminate the paper’s targeted meta-reasoning errors.

4. Missing citations. The paper mentions but does not cite: ReAct: Synergizing Reasoning and Acting in Language Models (https://arxiv.org/pdf/2210.03629.pdf)). Newer multi-step reasoning benchmarks such as LLM Reasoners: New Evaluation, Library, and Analysis of Step-by-Step Reasoning with Large Language Models (https://arxiv.org/abs/2404.05221)) should also be surveyed and cited.

**Questions:**

1. Could you provide a systematic error study separating meta-reasoning and object-level reasoning?

2. Could you report evaluation cost and time?

3. How does advance models such as GPT-5 perform on the benchmark?

---

> ### Author Response · Authors · 2025-11-24
>
> Hi, thank you for your review. We are grateful to read the points you raise as strengths of the paper, and hope to discuss the weaknesses and offer some clarifying points below.
> > 1. The benchmark relies on templated prompts and relatively narrow tools APIs, which is too simple compared to current practice, e.g., richer coding/spreadsheet/enterprise interfaces such as in SheetBrain: A Neuro-Symbolic Agent for Accurate Reasoning over Complex and Large Spreadsheets ([https://arxiv.org/abs/2510.19247](https://arxiv.org/abs/2510.19247)).
>
> Too simple for what, exactly? You say current practice, but this implies that the purpose of this work is to build the best QA system. The purpose is not to build a QA system, even over our domain, nor is the purpose to compete with other QA systems and show that our method is the best. We are not trying to maximise performance of LLMs on our dataset.
>
> The key element of our paper is the analysis of the reasoning behaviour of LLMs in greater detail, beyond final answer accuracy. In order to make such claims, we develop a dataset which, unlike any other that we are aware of, allows a more in-depth analysis of the actual reasoning process itself. This inevitably requires a trade-off, which in our case involved reducing the breadth in domain and increasing the depth of analysis possible. The experiments do not entail drawing very wide, sweeping claims and conclusions about all LLM/reasoning debates, and we do not make such claims in the paper. But, we do believe that our experimental setup, dataset and evaluation approach is enough to validate our claims about whether it is indicated that LLMs demonstrate a competent level of meta-level reasoning at the class of task that we embody.
>
> > 2. The study omits stronger/ newer models (e.g., GPT-5, Gemini 2.5 Pro) and larger mainstream open models (e.g., newer Qwen3 variants), making it hard to gauge the field’s current frontier.
>
> Yes, the study omits these models. However, evaluation of these models would make little difference to the general claim that we are trying to make about LLM-based reasoning. Newer models like GPT-5 would indeed likely perform very well, and better than many models that we evaluate. However, for the purpose of our study, that is, designing an evaluation method to analyse the meta-level reasoning of LLMs beyond simple final answer accuracy, the inclusion of these results would not impact the findings in a significant way.
>
> > 2. Insufficient error study. Beyond aggregate Acc./Precision/Recall and brief text, there is no systematic analysis of agent failure causes across models, nor full examples that illuminate the paper’s targeted meta-reasoning errors.
>
> We acknowledge that a more in-depth study of the errors made by the system is required, however, were suffering from a severe lack of space to discuss this in the paper.
>
> > 1. Could you provide a systematic error study separating meta-reasoning and object-level reasoning?
>
> The entire setup of the evaluation environment is based around separation of meta- and object-level reasoning. Object-level processes are outsourced to tools, which are error-free in their functionality. The only errors lie in the meta-level reasoning of the models.
>
> > 2. Could you report evaluation cost and time?
>
> If these are significant metrics which may affect the interpretation of our results, we can report these.
>
> > 3. How does advance models such as GPT-5 perform on the benchmark?
>
> We have addressed this above.
>
> Thank you again. We would politely invite reconsideration of our paper with respect to the extent to which our study has achieved its stated goals, bearing in mind the above points clarified about not designing a QA system, but an evaluation environment.

---

> > ### Comment · Reviewer_Qa2K · 2025-11-26
> >
> > Thank you for the rebuttal. Unfortunately, none of my original weaknesses were substantively addressed. In particular, the breadth and difficulty of the benchmark design directly determine the amount and credibility of the resulting insights. With the current limited domain, difficulty, and model set, the conclusions remain limited in scope—this is independent of whether the goal is to build the strongest QA system.
> > I recommend reading recent high-quality benchmark work and learning how to build a benchmark that meets top-conference standards—as well as how to write an effective rebuttal. Score remains.

---

> > > ### Author Response · Authors · 2025-11-26
> > >
> > > Hello, thanks for your response. In absence of the resources to create the large, broad and deep benchmarks that you reference, I attempted to shift focus on to the evaluation beyond final answer accuracy. And yes, I appreciate that I could not substantively address your weaknesses. Given their severity, e.g., breadth of evaluation, this is not something than can be easily addressed. I hope that I was at least able to explain a little more about the motivation and intended focus of the work.

---

### Official Review · Reviewer_nTSS · 2025-10-30

**Soundness:** 3
**Presentation:** 2
**Contribution:** 2
**Rating:** 2
**Confidence:** 3

**Summary:**

The paper distinguishes meta-level reasoning (planning steps and choosing tools) and object-level reasoning (executing retrieval and arithmetic steps), and introduces a tool-based multi-hop tabular QA benchmark from "World Bank indicators". Each question includes a set of "essential actions" used to evaluate models by their tool-call traces rather than final answers alone. Findings show that off-the-shelf LLMs often select appropriate tools, which means good meta-level reasoning, but may miss steps or hallucinate codes; providing 1/3-shot tool-use examples rarely improves accuracy, while error messages often help recovery; removing arithmetic tools significantly degrades performance, underscoring weak numeracy and the importance of external symbolic functions.

**Strengths:**

(1) The paper offers a deep exposition and evaluation of meta-level reasoning—specifically, a model’s abilities in task decomposition and task planning.

(2) Abstract concepts are described clearly, and the overall writing is fluent and easy to follow.

**Weaknesses:**

(1) A fundamental concern: in modern LLMs, it is difficult to cleanly separate “meta-level” and “object-level” reasoning in the symbolic-AI sense. For complex tasks in real world, most steps blend both forms of reasoning. The paper’s “first decompose, then execute” setup seems best suited to relatively simple multi-hop tasks (e.g., retrieval-augmented or calculation-augmented), and may not generalize to richer scenarios.

(2) The evaluation set is too narrow. Although the paper notes that the approach could extend to other domains, evidence based mainly on a word-bank dataset and a few simple tool-use tasks is insufficient to substantiate the broader claims.

(3) Formatting: Table 2 overflows the text bounds on the right and needs layout fixes.

**Questions:**

(1) When repeated calls with identical arguments are counted as one TP and the remainder as FPs, does this risk over-penalizing exploratory strategies that legitimately probe the tool state or recover from uncertainty?

(2) When keeping only retrieval and removing arithmetic tools, is the observed performance drop systematically related to model size or the reasoning mode?

(3) The chosen primary area “Datasets and Benchmarks” may not be a good fit?  This work reads more as an analytical/argumentative study than as a generally applicable, transferable benchmark.

---

> ### Author Response · Authors · 2025-11-24
>
> Hi, thank you for your review, we are grateful for your thoughtful comments and are pleased to read the points made in the strengths section. We wanted to discuss the weaknesses and offer some clarifying points.
>
> > (1) A fundamental concern: in modern LLMs, it is difficult to cleanly separate “meta-level” and “object-level” reasoning in the symbolic-AI sense. For complex tasks in real world, most steps blend both forms of reasoning. The paper’s “first decompose, then execute” setup seems best suited to relatively simple multi-hop tasks (e.g., retrieval-augmented or calculation-augmented), and may not generalize to richer scenarios.
>
> Yes, we acknowledge this difficulty, but we would say that this separation provides the primary motivation for our setup and is indeed a virtue of it: factoring out the object-level reasoning to explicit function calls (alleviating well-known weaknesses with respect to arithmetic), while allowing the model to generate free text plans (~meta-level reasoning). We are not claiming that this is not a solution for all branches of QA, especially QA focussed on natural language reasoning, but we do believe that our setup is a valid approach for a wide range of applications which require multiple steps of numeric fact retrieval and manipulation, and certainly not limited to our test 'World Bank geopolitical indicators' domain. We acknowledge that we did not offer concrete examples of this in our paper.
>
> > (2) The evaluation set is too narrow. Although the paper notes that the approach could extend to other domains, evidence based mainly on a word-bank dataset and a few simple tool-use tasks is insufficient to substantiate the broader claims.
>
> In our paper, we try to focus on examining one aspect of the 'reasoning' ability of LLMs in greater detail, rather than focussing on a very broad domain of questions and only relying on final answer accuracy. In order to make such claims, we developa dataset which, unlike any other that we are aware of, allows a more in-depth analysis of the actual reasoning process itself. This inevitably requires a trade-off, which in our case involved reducing the breadth in domain and increasing the depth of analysis possible. The experiments do not entail drawing very wide, sweeping claims and conclusions about all LLM/reasoning debates, and we do not make such claims in the paper. But, we do believe that our experimental setup, dataset and evaluation approach is enough to validate our claims about whether it is indicated that LLMs demonstrate a competent level of meta-level reasoning at the class of task that we embody.
>
> Thank you again, and thank you for your questions which we look forward to considering as the work evolves. We hope that our comments have clarified some of the errors in communication in our work, and we hope that this enables a different view of the work's virtues and scientific contribution.

---

### Official Review · Reviewer_sNQA · 2025-10-30

**Soundness:** 3
**Presentation:** 3
**Contribution:** 3
**Rating:** 6
**Confidence:** 4

**Summary:**

This paper introduces a novel question-answer benchmark that analyzes geopolitical information to evaluate the meta-level reasoning capabilities of LLMs. The questions require LLMs to decompose tasks into intermediate steps and apply data retrieval and mathematical operations. The dataset provides ground truth answers that include the essential actions needed to derive correct responses, enabling systematic evaluation of LLM performance.

**Strengths:**

1. The meta-level reasoning capability in the paper is defined as the high-level planning capability. The question design is highly consistent with this design goal. The proposed question construction format can be easily adapted to domains other than geopolitical.
2.  The dataset provides the correct sequence of essential actions, which enables assessment of partial correctness in LLM responses, thus offering a more nuanced evaluation of reasoning capabilities than binary success/failure metrics labels.
3.  The response post-processing method mitigates the influence of textual differences on semantically equivalent actions.

**Weaknesses:**

1. The task combines high-level planning with tool use, where LLMs need to understand tool call rules from the prompt.  LLM performance depends on both planning and tool-calling capabilities. Therefore, the evaluation may be influenced by confounding factors.
2. The presented metrics are precise and recall count for the content overlap between the LLM-generated action and ground-truth essential actions. For some cases, the action sequence is order-dependent, meaning different orderings of the same actions will lead to different results. Therefore, the precision and recall scores may not be robust

**Questions:**

1. I’m wondering, if providing paragraphs including both relevant and irrelevant information for the question (similar to HotpotQA), then LLM is asked to generate natural language inference steps to derive the final answer. Compared with the tool calling task, which setup provides a more precise estimation of the meta-level reasoning capability?

---

> ### Author Response · Authors · 2025-11-24
>
> Hello, thank you for your review. We'd just like to comment on some issues you raised.
> > 1. The task combines high-level planning with tool use, where LLMs need to understand tool call rules from the prompt. LLM performance depends on both planning and tool-calling capabilities. Therefore, the evaluation may be influenced by confounding factors.
>
> We incorporate elements of planning and tool-use into our evaluation, yes, and results do depend on both. In short, we aim to use tool use as a structured representation of the output of the planning (~meta-level reasoning) of LLMs, so the influence of a planning and tool-use component are not simply folded into one process from which each of their individual contributions to performance is not evaluable. Could you elaborate on what you mean by 'confounding factors'?
>
> > 2. The presented metrics are precise and recall count for the content overlap between the LLM-generated action and ground-truth essential actions. For some cases, the action sequence is order-dependent, meaning different orderings of the same actions will lead to different results. Therefore, the precision and recall scores may not be robust
>
> Order is important in some sense, yes. Let's say that if the LLM makes a `retrieve_value` call, prior to that we would expect to see calls like `search_for_indicator_names` and `get_country_code_from_name` to ensure the arguments are well-founded. However, the LLMs' behaviour is not such that it will make these calls *after* `retrieve_value` -- this is simply not something we would observe. Instead, it will simply not make these calls, which manifests as lowered precision and recall. So, yes, order is important when considering the task generally, but in our observation the behaviour of LLMs does bring about serious concerns about the robustness of the precision and recall evaluation.
>
> > 1. I’m wondering, if providing paragraphs including both relevant and irrelevant information for the question (similar to HotpotQA), then LLM is asked to generate natural language inference steps to derive the final answer. Compared with the tool calling task, which setup provides a more precise estimation of the meta-level reasoning capability?
>
> Thank you for this question. Yes, that's an interesting alternative evaluation scenario. The space of possible incorrect or irrelevant tool calls (which would demonstrate poor meta-level reasoning) is already very high granting significant potential for the LLM to perform poorly. Additionally, since our data is numeric, it's not immediately clear how to present meaningful quantities of adversarial input. If the data which the LLM had to incorporate into its answer was in natural language, this could be an interesting evaluation indeed.

---

### Meta-Review · Area_Chair_Rkos · 2026-01-08

**Summary:**

- The main concerns are summarized as follows:

1. Meta-level vs. Object-level Reasoning:
A fundamental concern raised by reviewers is the difficulty of cleanly separating meta-level and object-level reasoning in LLMs. While the paper makes this distinction, reviewers questioned whether this separation was valid for complex, real-world tasks, where reasoning steps often blend both forms.

2. Benchmark Design and Evaluation Scope:
Reviewers raised concerns about the narrow scope of the benchmark. The paper focuses primarily on a limited domain and simple tool-use tasks, which may limit the generalizability of the results. This focus on a narrow evaluation domain and tool set has led to doubts about the broader applicability of the claims made in the paper.

3. Methodology and Analysis:
The experimental methodology and error analysis were flagged as limited. Reviewers expressed concerns that the benchmark relies on prompt engineering with limited exploration of newer techniques. The lack of a detailed error analysis for distinguishing meta-level and object-level reasoning errors, as well as insufficient investigation into failure modes, were seen as weaknesses.

**Reviewer Concerns:**

- Concerns Still Outstanding:

1. Benchmark Design and Scope:
Despite the authors' defense of the benchmark's narrow scope, reviewers still questioned the generalizability of the findings. They noted that a more diverse benchmark, especially one that includes more complex tasks, would provide more robust insights into LLMs' meta-level reasoning capabilities.

2. Methodology: Prompt Engineering and Newer Frameworks:
The authors addressed the concern about the reliance on prompt engineering but did not adequately respond to the suggestion to explore more modern frameworks like the agentic framework or ToT. The rebuttal primarily defends the use of prompt engineering without directly engaging with the idea of incorporating these newer methodologies, leaving this issue unresolved.

**Reviewer Scores:**

None

---

### Decision · Program_Chairs · 2026-01-26

Reject